# Predicting Problematic Behavior in Autism Spectrum Disorder Using Medical History and Environmental Data

**DOI:** 10.3390/jpm13101513

**Published:** 2023-10-21

**Authors:** Jennifer Ferina, Melanie Kruger, Uwe Kruger, Daniel Ryan, Conor Anderson, Jenny Foster, Theresa Hamlin, Juergen Hahn

**Affiliations:** 1Department of Biomedical Engineering, Rensselaer Polytechnic Institute, Troy, NY 12180, USA; ferinj@rpi.edu (J.F.); krugeu@rpi.edu (U.K.); 2Center for Biotechnology and Interdisciplinary Studies, Rensselaer Polytechnic Institute, Troy, NY 12180, USA; krugem@rpi.edu; 3Department of Mechanical, Aerospace, and Nuclear Engineering, Rensselaer Polytechnic Institute, Troy, NY 12180, USA; 4The Center for Discovery, Harris, NY 12742, USA; daniel.robert.ryan@live.com (D.R.); canderson@tcfd.org (C.A.); jenny.foster@tcfd.org (J.F.); thamlin@tcfd.org (T.H.); 5Department of Chemical and Biological Engineering, Rensselaer Polytechnic Institute, Troy, NY 12180, USA

**Keywords:** autism spectrum disorder, challenging behavior, machine learning

## Abstract

Autism spectrum disorder (ASD), characterized by social, communication, and behavioral abnormalities, affects 1 in 36 children according to the CDC. Several co-occurring conditions are often associated with ASD, including sleep and immune disorders and gastrointestinal (GI) problems. ASD is also associated with sensory sensitivities. Some individuals with ASD exhibit episodes of challenging behaviors that can endanger themselves or others, including aggression and self-injurious behavior (SIB). In this work, we explored the use of artificial intelligence models to predict behavior episodes based on past data of co-occurring conditions and environmental factors for 80 individuals in a residential setting. We found that our models predict occurrences of behavior and non-behavior with accuracies as high as 90% for some individuals, and that environmental, as well as gastrointestinal, factors are notable predictors across the population examined. While more work is needed to examine the underlying connections between the factors and the behaviors, having reasonably accurate predictions for behaviors has the potential to improve the quality of life of some individuals with ASD.

## 1. Introduction

Autism spectrum disorder (ASD) is a neurodevelopmental condition that is defined by difficulty in communication, social interaction, and restricted repetitive behaviors. ASD is estimated to affect about 1 in 36 children in the United States by age 8 according to the CDC [1]. Despite being categorized and diagnosed by a set of behavioral criteria [2,3], ASD is known to often be associated with several co-occurring conditions that affect a multitude of physiological systems [4,5]. Three major areas of ASD-associated comorbidities include sleep disorders [6], gastrointestinal (GI) problems [7], and immune disorders [8]. These comorbidities have the potential to cause an individual pain or discomfort, which may affect their behavior. While many atypical behaviors are associated with ASD, the focus of this paper is on two behaviors that can pose danger to the affected individual and/or others around them: aggression and self-injury. Aggressive behavior can have severe consequences for the individual, their peers, and caregivers [9,10], while self-injurious behavior (SIB) can cause serious harm to the individual [11].

Aggression has been shown to be associated with sleep abnormalities [12,13,14]. Aggression is also associated with several other behaviors, including SIB [14,15], ritualistic, and sameness behaviors [15], is more likely to occur at younger ages [15], and is also associated with gastrointestinal problems [14]. SIB is also a highly prevalent behavior in ASD; in one study, 50% of participants expressed SIB [16,17]. SIB has been shown to be associated with comorbidities of ASD [18], including sleep abnormalities [13,14]. Low expressive and receptive communication levels were strongly associated with lower levels of aggressive behavior and somewhat associated with lower levels of SIB in infants and toddlers in one study [19]; however, other studies on older children or adults have found the opposite. Duerden and colleagues found that abnormal sensory processing, a need for sameness, IQ, and abnormal social functioning were associated with self-injury in children and adolescents [20]. In adults with ASD but without intellectual disability (ID), self-injury in over half of participants was reported to help with low- (e.g., depression-associated) or high-pressure (e.g., anxiety-associated) affective imbalance; additionally, alexithymia (difficulty verbally communicating emotions), depression, anxiety, and sensory sensitivity were all varying when comparing individuals who currently exhibit SIB and those that never do [21,22]. The consequences of this behavior can be particularly severe; even though it is directed to self-stimulate or for social reinforcement [11,23], it can lead to injuries and it is associated with suicide [22]. Both of these behaviors have been shown to be associated with overt emotional dysregulation; however, this is highly heterogeneous to individuals. Additionally, individuals appeared to be in distress post-behavior more often than prior, demonstrating the possible negative impacts of these behaviors [24].

For both mothers and fathers, more frequent challenging behaviors of their children are associated with the frequency and intensity of daily stressors, as well as increased depressive symptoms [25]. Aggression is significantly associated with parent/child and medical/legal functional impacts; SIB is associated with a lower quality of life for the child and their family, as well as parent/child and medical/legal functional impacts [26,27]. Both aggression and SIB are in the top five predictors for psychiatric hospitalization in children with ASD, with aggression being the strongest predictor [28]. While pain assessment in individuals with ID and ASD is a relatively new field, a recent study found associations between pain and challenging behavior [29]. Courtemanche and Black found that parents perceived pain to be lower as profundity of ASD increased, even though there were no significant differences in pain expression between children with ASD and typically developing (TD) children [30]. Individuals experience higher frequencies of challenging behavior on sick days vs. well days, indicating that pain and discomfort impact behavior [31]. These behaviors, which may be expressions of pain, not only harm the individuals that exhibit them, they also increase stress for caregivers; the ability to prepare on a day when a behavior will occur may help to alleviate some of the stress and potentially reduce the harms of the behavior.

While the risk factors of having severe behaviors have been extensively examined [16], predicting the likelihood that a behavior will occur on a certain day is less studied. However, from a caregiver’s perspective, knowing if someone is at an increased or decreased likelihood of exhibiting behaviors would be very beneficial. As the consequences of these behaviors may cause serious harm to individuals, peers, or caregivers, allowing the caregiver to know in advance which individuals to accommodate would be beneficial. Days can be planned around whether a challenging behavior is expected to occur; for example, delaying a doctor visit if that is already known to cause stress to the individual. Planning a schedule around the times where a student was likely to self-injure allowed them to be more engaged in the classroom for one student with ASD [32]; and in general, behavior intervention techniques such as antecedent manipulation can reduce the frequency of challenging behavior over time in the classroom setting [33]. On average, challenging behavior interventions are effective for those with ID, while the degree of effect highly varies between individuals [34]. There is no federal minimum requirement in the United States for staff:resident ratios at long-term care facilities [35]; thus, staff members may have to supervise several residents at once. Adjusting the schedule and/or group in advance may lessen the harmful impacts of the behavior.

Physiological and motion indicators, measured via a wearable biosensor, have been shown to predict ASD-associated aggression [36]. However, these predictions of aggression only happen approximately one minute before an episode occurs. While this finding is very useful for potentially intervening to minimize harm to the individual and others, a more long-term prediction would be desirable for residential facilities. For example, this could result in ensuring additional supervision for someone who is more likely to have a behavioral episode that day. Similarly, SIB has been predicted using accelerometer data with a high degree of accuracy; however, this approach provides detection while the SIB event is occurring, rather than giving prior warning [37]. Alivar, Carlson, and colleagues [38,39,40] have completed several pilot studies using bed sensors for next-day behavior prediction, but these studies had a small sample size. In a study by Cohen et al. [13], sleep-derived variables over a period of the previous two weeks were shown to predict behavioral episodes of several challenging behavior types, including aggression and SIB for participants with profound ASD at two residential facilities. While these biosensors and motion sensors appear to be well-tolerated [36,37], this may not be applicable to all individuals with ASD and high support needs; the sample sizes of most of these studies are small and given the heterogeneity of the ASD population, there will almost certainly be a subgroup that cannot tolerate constant sensor wear. Given the state of research in this field, there is clearly a need for more investigations that focus on large sample sizes and data that have been collected over extensive periods of time for predicting undesired behaviors.

Research in ASD often focuses on populations that require lower support needs [41,42,43,44]. Focusing on the underserved population with profound autism [45] helps to inform care for individuals that have behaviors with severe outcomes and may stand to benefit most from determining the possible causes of the behavior. Additionally, any knowledge about factors affecting undesired behaviors, as well as any predictions about increases or decreases of the likelihood of undesired behaviors, can have a significant impact for minimizing the harms associated with these behaviors. Although behaviors may be associated with discomfort from comorbidities [14,18], many individuals with ASD and high support needs may not be able to communicate this verbally, and assessments of behavior function are often based on observed symptoms from a psychological perspective rather than a physiological one [46]. Therefore, investigating these factors may help to better meet these individuals’ needs and improve their quality of life. 

This work will make use of an almost unique dataset which combines information over extended periods of time with very detailed daily medical information and also environmental data for its analysis and predictions. These data have not been used in any prior work and the raw data are available as part of this work. Making use of this dataset, this work seeks to predict the occurrence of aggressive behavior and SIB on any given day from data collected about the individuals on a prior day.

## 2. Materials and Methods

### 2.1. Cohort Selection and Data Inclusion Criteria

Data from The Center for Discovery (TCFD) in Harris, New York, were used for 566 individuals who lived at the center during the time period of July 2015 through November 2021. The data included in this analysis were selected from an archive of data that were routinely collected as part of ongoing clinical care and monitoring at TCFD. TCFD has an Institutional Review Board (IRB)-approved policy that their residents’ data can be used in a de-identified state for research studies to better understand and predict program effectiveness. All data used in this study were fully de-identified prior to cohort selection and eventual analysis. As such, IRB approval was not required for this work.

The data format was a time series dataset for each participant, recording sleep, GI, and behavioral data on each day. After a person was admitted to TCFD, a period of one month was allowed to ensure that the individual has settled into new routines and also that data collection began for behavior, sleep, and GI data. If the data collection did not start for one or more of the categories, the latest date across the first record of behavior, sleep, GI data, or the one-month period was used as the start of data collection. First, the data were filtered (Figure 1) such that each individual resided at TCFD, to ensure accurate and consistent data collection by the TCFD staff. The dataset was then filtered to only include persons less than 19 years of age at the start of the study, which limited the study to adolescents. A diagnosis of ASD, confirmed by professionals at TCFD, was also required. Up to 18 months of data were used for each person and the data were required to have at least 20 data points. There was a maximum class imbalance threshold of 90%, either examples of behavior or of no behavior. Missing data occurred from leaves of absence (LOAs), when the individual was away, visiting family, or when data were unrecorded for that period. These criteria are similar to those used in the study by Cohen et al. [13]. After applying all exclusion criteria, a final dataset of 80 individuals was used for the remainder of the study. These 80 individuals included those with a sufficient number of episodes of aggressive behavior, self-injury, or both combined for the analysis. It should be noted that there was considerable overlap between the challenging behavior types, but some were excluded from other classes; for example, if an individual had 80% aggressive behavior and self-injured the remaining 20% of the period, they were included in SIB and AGG but not BOTH cohorts, as the class imbalance for BOTH would be over 90%.

Limited demographic information was available on the individuals. In all cohorts, both ends of the age range were represented (Table 1). Typically, males have four-times-higher rate of ASD diagnosis than females [1], and a high male to female ratio was also present in our cohorts.

### 2.2. Details of Longitudinal Data Used

#### 2.2.1. Data on Exhibited Behaviors

Challenging behavior is often performed to serve a function; for example, functional behavior assessments (FBAs) can be completed to determine if a problem behavior is performed to obtain attention or a desired object. A review of the FBAs literature found that in students with ASD or ID, challenging behavior was most often used as a means of escape [47]. Functional behavior analysis (FBAn) can be performed by experts, as part of an FBAs, to determine the function of a behavior as well, by systematically perturbing conditions and observing behaviors to determine the aim of the problem behavior [48]. FBAn research has found that there are subtypes in functions of problem behavior, which necessitate an examination of biomarkers [49] and by extension, we examine the role of physiological and environmental variables. In some cases, however, if the behavior is severe and dangerous, an FBAn may be too risky to include as part of an FBAs. Every individual admitted to TCFD presenting with significant challenging behavior receives applied behavior analysis-based behavioral services overseen by a board-certified behavior analyst and licensed clinical psychologist. Behavioral staff perform FBAs for each targeted behavior in accordance with regulatory requirements, and develop individual behavior intervention plans (BIPs) as needed which address both skill acquisition and behavior reduction procedures related to the function of behavior. All the records involved in the present study are derived from data collection pertaining to specific behavior intervention plans. However, the clinical consultants did not include behavioral function for variable assessment because many individual behaviors are characterized by multiple factors [50,51,52,53].

At TCFD, direct care staff are required to collect behavior data for individuals who have been clinically identified as requiring formal behavior intervention. Behaviors targeted for intervention are specified within the BIP, which includes detailed information on how to identify their occurrence and how to document. All behaviors were documented across three shifts covering a 24 h period at TCFD, from 7 a.m.–3 p.m., 3 p.m.–11 p.m., and 11 p.m.–7 a.m. the next day. The behaviors documented within the cohort were highly varied. In consultation with lead behavioral staff, each behavior was categorized as aggression, SIB, or neither of the two. If there was a behavior during at least one recorded shift of a day, it was considered that there was a behavior that day. Similarly, if there were no behaviors during any recorded shifts of a day, records of the individual on that day were coded as no behaviors. If leaves of absence or missing data comprised any recorded shifts that day, the day was excluded from the analysis. For some individuals, there were additional fine-grained details including the intensity and duration of behavioral episodes. If any information regarding the episodes was provided, this was considered a valid behavior instance. However, if the data collection was performed using time sampling (the percent of time that the behavior occurred) for any shift of a day, the entire day was excluded due to different data collection methods. The binary behavior data are shown in Figure 2 for each individual. It should be noted that the persons in the three subfigures with the same label numbers do not correspond to each other as numbering was performed separately for each category.

#### 2.2.2. Sleep Variables

Sleep duration and quality has been extensively studied for ASD [6,54,55,56] and as such was one of the variables investigated here. Sleep data have been found in several studies to be a predictor of challenging behaviors [13,38,39,40,54]. In particular, sleep onset and sleep duration were found to have the strongest negative associations with behavior, while sleep onset and efficiency variabilities had the strongest positive associations with challenging behavior, across the features measured by Cohen and colleagues [13]. Sleep interruptions were found to have a negative association with challenging behavior as well, but the association with behavior was less strong across the different numbers of prior nights measured [13]. Individuals in a non-ASD pediatric population with ID and behavioral disorders had a significantly lower night sleep duration than TD individuals as age increased [57]. Among adults with ID, those who exhibited SIB also had shorter sleep durations than those who did not [58]. Overall, poor sleep is associated with anger [59,60] and emotion dysregulation, which may lead to problem behavior. The collected sleep data included the sleep duration in number of hours, and the number of sleep interruptions each night. Data were cleaned such that sleep hours were truncated to be between 0 h and 24 h. Sleep hours and interruptions were recorded via bed checks each night at approximately hourly intervals.

#### 2.2.3. Gastrointestinal Variables

There have been several studies related to GI symptoms and ASD [7,14,61] which makes investigation of GI symptoms one of the key variables studied here. While GI problems are associated with some ASD behavioral characteristics and with sleep abnormalities, there was not a significant association observed between ASD with or without GI problems and the presence of aggression or SIB in one study [62]. Interestingly, GI problems were associated with sleep abnormalities in the same study [62]. Microbiota transfer therapy has also been shown to reduce symptoms of problem behavior post-treatment in children with ASD, in addition to reducing symptoms of GI pain [63]; however, this investigation was performed on a weekly basis rather than daily. Thus, an investigation of whether pain from GI symptoms is correlated with challenging behavior is warranted. GI data were recorded by staff, nurses, hospital staff, parents, or on occasion, self-reported. Staff, nurse, hospital, and self-reports were all retained, while parent reports were not, as it was assumed that the individual was absent from TCFD in that case. GI data concerned each person’s bowel movement(s) (BMs), or lack thereof, on each day. A valid bowel movement was considered to have either size or Bristol scale information, or had to be recorded in a note by staff. Size information was not further used, as it was qualitatively recorded. The Bristol scale [64] is a qualitative but standardized scale to visually assess BMs, ranging from 1 (constipation) to 7 (diarrhea). A Bristol score of 3–4 is ideal. If there were no BMs that day, the Bristol score was coded as 0. The Bristol score was averaged if there were multiple BMs in a day, and the maximum and minimum Bristol of each day were also retained. Each valid BM on each day was counted and the total amount of BMs was recorded. Because BMs were sometimes only recorded if there was an event, a buffer period of 3 days was added. If there were over 3 days with observations but no BMs, and the number of days without BM was not indicated, subsequent days were considered as missing.

If an individual was constipated, interventions were administered in the form of laxatives or other medications to encourage a BM, sometimes including prune or fluid intake. All interventions were coded into one binary variable stating whether or not an intervention was given that day. The presence of interventions was notably sparse; however, this is in part due to the nature of the variable. Additionally, a variable indicating menstruation was included with the gastrointestinal variables.

#### 2.2.4. Environment Variables

The environment has been shown to affect challenging behavior in children living in residential facilities [65]. That being said, the impacts of the external environment in terms of quantitative natural phenomena and their effect on challenging behavior have been less studied. To retain data quality and prioritize data collection related to patient care, environmental variables were obtained from external sources.

#### 2.2.5. Allergen

In order to obtain data for this study, allergen data were gathered from the American Academy of Allergy, Asthma, and Immunology (AAAAI)’s [66] Armonk, NY [67], station, which was the closest active station to TCFD. Due to the sparseness of each allergen species, the AAAAI-designated categories of tree, grass, and weed pollen were used. A total including tree, grass, weed, and unidentified pollen was also included. All days prior to the first recorded observation of the year were filled as zeros. Because the station did not record data every day, missing observations during allergen season were filled with the previous valid observation until the next recorded value or the last observation of the season. After the last observation was recorded, all pollen levels were recorded as zero for the remainder of the year. 

#### 2.2.6. Weather

Weather data were obtained from the NOAA [68,69]. A study by Bolton et al. suggested an effect of weather on some characteristics of individuals with ASD [70]; another study suggested minimal, if any, effects and that weather and behavior in ASD warranted further investigation [71]. The weather data collected included maximum temperature and minimum temperature (degrees F), precipitation level, snowfall, and snow depth (all in inches), i.e., the five core values recorded by the NOAA. The closest station to TCFD with high record completeness for these core values was the Rock Hill 3 SW, NY, station, which is located approximately 20 km from TCFD. If any data quality flags were raised by the NOAA, the observation for that day was discarded. 

#### 2.2.7. Lunar Cycle

While the inclusion of lunar data is controversial, there is evidence that the lunar cycle affects sleep quality and melatonin [72]. Moon data, thus, may be regarded as a proxy for other, unobserved sleep variables. Lunar data were obtained from MoonCalc.org [73] at the precise latitude and longitude of TCFD. Lunar data included moon rise and moon set, which are typically the times that the edge of the moon is first seen over the horizon and may not happen on a given day [74]. In this case, moon rise and moon set were taken when the upper limb of the moon touched the horizon [73]. Moon culmination was recorded as when the moon crossed the meridian [73]. Sometimes, the moon rise and moon set did not happen on a given day. In those cases, if moon rise was before midnight on that day (did not occur) it was recorded as midnight, and if moon set was after 23:59 on a given day (did not occur), it was recorded as 23:59:59 to avoid missing data. All times were converted to percentages of a 24 h clock during data cleaning. The distance from the centers of the earth and the moon were recorded in kilometers, the altitude of the moon was calculated as the angle between the center of the moon and the horizon accounting for refraction in degrees, and the moon azimuth was the angle taken between the earth at the meridian plane and the moon at the vertical plane, in degrees [73]. The length of the shadow of the moon was taken in meters relative to a one-meter-tall object [73]; in the event where there was no shadow (during the waxing phases of the moon and new moons), the shadow length was changed to 0 m during the data cleaning process. The moon phase was recorded as well; because categorical moon phase and continuous illumination percentage were recorded in the same variable [73], only the percentage was used as it was more precise. The moon age was recorded as the number of days since the last new moon out of a lunar cycle [73] and was converted to a percentage. The date and time of the next new and full moons were also recorded [73] and this variable was converted to the number of days from the current date to the next new and full moons in whole days, with time not accounted for. Perigee and apogee data, the datetime of the next event the moon was closest and farthest from Earth, respectively, were available [73], but were not under the core data section and so were not used.

To ensure the consistency of data collection with behavior shifts, the moon data were taken at 7 a.m. each day. The timing of data collection would not change the moon rise, set, or azimuth variables on each day but would change the angle and distance variables. The data collection timing is shown in Figure 3.

### 2.3. Feature Engineering and Variable Selection

Preliminary analysis showed that many of the variables had a minimal association with the behavior response variable(s) at the population level. This was not surprising given the large heterogeneity of behaviors and medical factors found among individuals with ASD. Instead, individual analysis was performed, along with feature engineering. Constipation and diarrhea variables were created where constipation included having a Bristol score of below 3, having no BM that day, or requiring a BM intervention that day. Diarrhea was indicated by having over 3 BMs in a day, or having a Bristol score of above 5. An abnormal BM variable was also created, which included a Bristol score on either the constipation or diarrhea side, having more than 3 BMs or having no BMs in the day, or having an intervention that day. The most extreme Bristol difference was calculated as well, from the optimal Bristol point of 3.5. All variables, both categorical and continuous, were z-score standardized for the population. The complete list of considered variables can be found in Table A1 of Appendix B.

### 2.4. Modeling Methods, Model Validation and Model Selection

The construction of the classification models relied on the adaptive linear neuron (ADALINE) [75], a single-layer neural network that is able to adjust its weights as it trains by comparing the error from the prediction with the outcome. The network inputs were features constructed by a direct kernel transformation of the original variables discussed above. A direct kernel transformation is based on the squared length of the difference between a data point and a prototype, which is one randomly selected data point. The squared length is then divided by a kernel parameter to form the argument for the exponential function, i.e.,
(1)$$f_{ij}=\exp(-|\left|\;x_{i}-p_{j}\;\right|{|}^{2}\;/\;\sigma )$$
where $x_{i}$ is the *i^th^* data vector storing the values of the original variables, $p_{j}\;$ is the *j^th^* prototype (randomly selected from the entire set), σ is the kernel parameter (scaling factor) and $f_{ij}$ is the *j^th^* feature variable for the *i^th^* data vector. Applying a variety of different numbers of prototypes, it was found that 15 prototypes resulted in the best cumulative performance when evaluated for each of the 80 participants, i.e., 1 ≤ j ≤ 15. The output of the ADALINE, i.e., the linear combination of these 15 features plus a bias term, was passed through a logistic activation function, which yielded a probability value between 0 and 1 to express how likely a participant is expected to exhibit the undesired AGG, SIB, or BOTH behaviors. The parameters of the single-layer ADALINE network architecture were trained by applying the Widrow–Hoff rule for a total of 1000 epochs based on a random split of the data for each participant into 85% of training and 15% of validation data. In addition to that, 30 different random splits were applied to guarantee that the generalizability of the model structure applied was rigorously tested. The results for all splits were retained for further analysis. 

Note that some validation sets did not contain data vectors from both classes, particularly non-normal behavior. These were excluded from the subsequent analysis. In addition to the direct kernel transformation, the analysis herein also includes the use of logistic regression classifiers as a benchmark. Given the imbalance in the number of occasions where normal to non-normal, i.e., AGG, SIB, or BOTH, behaviors were recorded, the balanced classification rate, or BCR, was used as a classification metric to select models. This was to ensure that low sensitivity but high specificity percentages did not produce high accuracies.

It is also of interest to identify which of the original variables are important in discriminating between undesired and normal behavior and which variables have a minor but noticeable to negligible contribution. For this, we applied a standard approach in that we simulated 1000 values for one variable that are drawn from a standard normal distribution and assigned values of 0 to all remaining variables. Recall that the variables were standardized and, thus, have a mean of 0 and a variance of 1 after standardization, i.e., the 1000 simulated values cover a comparable range to the original standardized variables. The process of “exciting” a single variable only, while the remaining ones are kept constant to 0, starts with the first variable, where we record the standard deviation of the network output. Each variable is “excited” once, which results in one calculated standard deviation for each variable. Finally, these standard deviations for each feature were scaled such that the range of each standard deviation was between 0 and 8. Guided by the “Pareto 80-20 rule” [76], we consider any scaled standard deviation that assumes a value greater than 2 as a significant contributor, values below 0.5 as negligible, and values between 0.5 and 2 as indicating variables that have a noticeable but minor contribution. We employ a stricter cutoff and plot any values greater than 4 as significant contributors.

### 2.5. Reproducibility

The classification methods were implemented in MATLAB. The random seeds were generated using integers 1–30 and the Threefry algorithm. The model selection code was run in MATLAB R2020a using a high-performance computing cluster. The analysis code was run on a Dell Inspiron 15 in MATLAB R2020a. All modeling and analysis code as well as the data files input into the models can be found at the GitHub link we provide in the Appendix A and Data Availability sections.

From the raw data, data analysis was implemented using Python, and the Sankey diagram and circle plot were also generated in Python.

## 3. Results

### 3.1. Data Pre-Processing and Results from Sensitivity Analysis and Data Reduction

Given the large number of input variables, a detailed sensitivity analysis was performed to determine the key factors retained for further investigation. One goal of this analysis was to retain at least one feature for each of the five major categories (BM, sleep, allergies, weather, and moon). Since no allergy information was available for the 80 individuals of this study, i.e., who was allergic to what type of pollen, only the total allergen data was selected as the representative variable from the allergen group. Similarly, due to correlations among the weather variables, only the maximum temperature was selected as the representative variable for the weather data, due to its ability to impact individuals when they went outside as opposed to spending time indoors in a temperature-controlled environment. While an extensive amount of lunar information was available, many of the variables were correlated and/or showed no significant effect on the outcome. As such, the moon phase variable (percentage illumination) was selected as representative due to its correlation with sleep hours. All analysis shown below made use of this reduced dataset as including more variables did not result in an increase in prediction accuracy.

### 3.2. Linear Analysis Using Original Variables

The presence or absence of behaviors for individuals were predicted and the accuracy of the predictions was assessed, where each individual used a different model, including different features and weights, for the predictions. The large heterogeneity found among the population resulted in poor predictions at the population level, therefore requiring individual models for each person. Linear methods were explored first, with Figure 4 showing the results for all individuals. The median performance of the linear methods was close to 50%, necessitating the exploration of nonlinear methods.

### 3.3. Nonlinear Analysis Using Direct Kernel Transformation

The analysis was repeated with a direct kernel transformation and a logistic activation function, and the results are shown in Figure 4. The nonlinear results increase BCR substantially, with most individuals having a median BCR above 50%, and a few approaching 90%. The average sensitivity and specificity of the models were also examined versus the proportion of days of behavior in the test set, shown in Figure 5. Models showed improved sensitivity when the percentage of days with behavior was >50%, while models showed improved specificity when the percentage of behavior was <50%; these findings suggest that there is an improved performance for individuals with less class imbalance.

Overall, the nonlinear results showed a significant performance improvement relative to the linear results. However, the use of all features in the model does not provide an indication as to which factors are the most important for prediction. While sensitivity analysis indicated a unique profile of feature importance for every individual, a summary of the features across the population is shown in Figure 6.

To further characterize the correlations between variables, hierarchical clustering was performed on the individuals where the balanced accuracy exceeded 80% (Figure 7). Many BM variables were correlated with each other in two distinct clusters for all cohorts. The third cluster included sleep, moon, weather, and menses variables, and the BM intervention variable in the case of BOTH.

## 4. Discussion

From a clinical perspective, a balanced accuracy exceeding 80% for the predictions is generally considered the minimum number to follow up on any predictions in practice. Using the individual models, it was possible to predict the occurrence of a behavior on the next day from the prior day’s data for 15 individuals of the 70 in the AGG cohort, 6 of 39 in the SIB cohort, and 16 of the 72 individuals in the BOTH cohort (Figure 5). According to this, approximately 15–20% of the cohort reached this 80% prediction accuracy threshold, indicating that the presented approach should be further studied for a subgroup of individuals exhibiting aggression and/or SIB where the balanced accuracy exceeds this threshold.

Aside from the 20% of individuals where the predictions exceeded a clinically relevant accuracy threshold, the models for all individuals exceeded 50% prediction accuracy. While these other models may not be useful in clinical practice, they nevertheless show that the variables investigated here, in particular GI variables exhibit some correlation with the presence/absence of unwanted behaviors the next day. 

It is clear from this study that what leads to an individual exhibiting a challenging behavior cannot universally be predicted from just one variable set and instead requires a number of different factors that have to be taken into account and furthermore, the best prediction model varies from individual to individual. While population models were also developed, they had a poor prediction accuracy and the results were not included in this work. As far as the individual factors are concerned, GI variables were prominently present in most individual models, while this was not always the case for variables from the other groups. That being said, models that were just trained on GI variables did not perform nearly as well as the models that included variables from all factors discussed here (results not shown), indicating that while GI variables were a significant contributor to the presence/absence of behavior the next day, they were not the only factor. This is also apparent from the dendrograms shown in Figure 7. While GI variables are represented prominently, both because of their importance but also because there are many of them, including five or more variables always includes some non-GI related variables indicating that these other variables contain information that complements the information found in the GI variables.

As far as other variables are concerned, the lunar illumination percentage was not shown to be significant for all individuals, but it was a significant predictor for some. Also, there was a correlation between sleep duration and the moon phase which can be seen in Figure 7 as these two variables appear in the same cluster. Anecdotally, a full moon is associated with poorer sleep; however, studies have shown mixed results [77,78]. However, only sleep duration and interruptions were measured in this work, and other, unobserved sleep measures may also show correlations with the moon data. In general, including the moon data did not improve model predictions as long as sleep variables were included as important predictors, indicating some form of correlation between the two types of variables. Menses was a significant predictor for the SIB cohort, and this is consistent with findings in the SIB literature [79,80].

Weather data were also, overall, important for predicting behavior. While a characteristic of ASD is increased sensory sensitivity, results in the literature on sensitivity to weather have been mixed. One small study suggested that weather salience is higher in those with ASD [70], while another small study has measured weather variables versus behavior episodes each day and found few, if any, relationships [71]. In our study, we found weather to be an important variable. The effect of weather on individuals can take several forms and these include, but are not limited to, feeling uncomfortable due to particularly hot or cold weather or changes in their daily schedule due to extreme weather conditions. The ultimate cause of this correlation can unfortunately not be determined from the data that we have. More studies are needed to examine the effect of weather on discomfort in individuals with ASD, and could also include examining the effects of pressure or humidity. 

GI problems are common among children with ASD [81]; however, they were not shown to be significantly associated with aggressive behavior compared to children with ASD and no GI problems in a previous study [82]. Our results showed BM variables were significant predictors of behavior for nearly all individuals, and we have noticed correlations between the BM variables as well. It can be seen from Figure 7 that there are three significant clusters of variables for the predictions of all three groups (AGG, SIB, BOTH), two of which exclusively contain BM variables, while the third cluster contains everything else. Mazefsky and colleagues have found an association between GI affective problems, but not with aggressive behavior [83]. Maenner and coworkers found GI problems not to be significantly associated with aggression or SIB, but did find them to be associated with oppositional behaviors [62]. Additional analyses are needed on a time series basis to determine if GI problems impact behavior, as GI-associated pain is a concern in this population. That being said, the fact that GI variables are important predictors for behavior in most individuals indicates that a correlation exists for the individuals included in this study.

The total allergen variable was overall not shown to be as predictive as the variables of most other categories. The effect of the presence of allergens on individuals can depend on a number of variables not investigated here, such as if someone is taking allergy medication, or how often or how long each individual goes outside. Additionally, it should be noted that this variable is seasonal, and therefore at constant low levels for several months of the year, which may eliminate it from consideration in some individuals. Additionally, each person may have a unique allergen profile that was not captured by the pollen data. Many species appear sparsely, and so were combined into a summed total of the general trees, grasses, and weeds categories, but individuals affected by only some species may not be accurately represented by this aggregation. One study showed that individuals who engaged in either SIB, aggression, or both had reactions to foods, suggesting that food allergy/sensitivity may increase the likelihood that an individual may exhibit one or both of these behaviors; however, it is important to note that this finding is in general, not on a day-by-day basis [84]. Another study found a link between allergy symptoms and problem behavior; however, only one individual with allergy symptoms was evaluated [85]. A consensus report of physicians recommended examining the association between allergens, GI, and behavioral symptoms in patients with ASD [86].

A detailed comparison of our findings with those reported by Cohen et al. [13] suggested some agreement regarding the relationship between sleep and behavior. While we also observed that sleep could predict behavior modestly on its own using our nonlinear methods, our predictions improved significantly when sleep data were combined with GI data. This is in contrast to what has been found by Cohen [13]; in our case, the sleep variables were not the most promising predictors of behavior on their own. This difference may be due to the less detailed sleep data available at the time of collection; onset and offset data were not available for the majority of the dataset, leaving only duration and interruptions. However, in the model combined with other variables, sleep was a strong predictor after GI for individual behavior types. The link between sleep and behavior should still continue to be examined with wearables or bed sensors, both of which have been shown to be promising [38,39,40,54]. Poor sleep is associated with increased challenging behaviors; however, only night awakenings were associated with SIB on a daily basis in a study by Abel et al. [54]. As we did not find a significant relationship between night awakenings and challenging behavior, sleep metrics and behavior should be further studied. 

Both linear and nonlinear analyses were performed for this work. When developing nonlinear models for individuals, the median accuracy of the prediction increased significantly, but this was very dependent on the individual with some scoring as high as 85% while others stayed slightly, but notably, above 50%. Also, we found that the individual models were highly heterogeneous, but that some variable subsets, including GI, sleep, and weather variables, tended to have a higher predictive accuracy. However, this finding was mainly limited to individuals who did not have a large class imbalance, i.e., those where behaviors were either absent or present in less than 70% of all days. When the class imbalance was above 70%, i.e., for >70% of all days there was no behavior or almost always a behavior, then the advantage of a predictive model using these inputs disappeared for most individuals, and the individuals’ behavior could be more reliably predicted by analyzing how often someone, on average, has had behaviors in the past. The exact reason behind this observation is unclear, but the lack of data, i.e., if an individual exhibits behaviors on most days then there are few days with no behavior that the model can be trained on and vice versa, is definitely one of the potential reasons underlying this observation.

From our individual models, we found our prediction accuracies to be in a similar range, but on average higher, than those found in the current literature [13]. In particular, GI-related variables show an association with behavior and prediction accuracy which can be further improved when also including sleep variables. Future work will test these predictions on new data from these individuals and also will investigate why the prediction for some individuals is much better than for others.

Lastly, the choice of this particular dataset for this study needs to be discussed. Most clinical studies involve data collected from specific clinical trials where individuals are recruited for the study, whereas our work involves data collected as part of routine clinical care for individuals participating in a residential educational program. Regular clinical trial data tend to be cleaner than data collected as residents go about their daily lives; however, the data used here are more reflective of what one would find in practice. Lastly, individuals with profound autism are generally underrepresented in clinical studies [41,42,43,44] and as such a study focusing on challenging behaviors in ASD should make use of the most relevant data available for this purpose. 

Data have been additionally included from outside sources, with locations chosen to minimize missing data and distance from TCFD, but were not collected at the same place or by the same individuals and there may be some variability of environmental conditions. Inclusion of these environmental data, however, allow data that are collected consistently and externally to be leveraged which could assist with further automating the models when implementing in clinical practice.

On the individual level, we use random 85:15 splits of training and testing data even though this is a time series problem due to data limitations preventing blocked cross-validation. However, a prior study found that there were no consequences associated with using 5-fold cross-validation as compared to blocked cross-validation for stationary time series [87]. To ensure the generalizability of the model, we performed 30 random splits of the data and reported the mean and median results.

Opportunities to expand this work include implementing wearable technology, examining the directionality of prediction, and expanding the diversity of the sample. Wearables and bed sensors are useful tools to detect behavior; however, the access to this technology must be considered. This includes data privacy and comfort concerns of the individuals and their families. The relationship between prior data and behavior may also be bidirectional and should be further explored. Performing the analysis on an expanded sample will allow the resulting models to be more generalized and robust.

### Limitations

As with any study involving real-world data, there are several limitations to this study. We do not have multiple observers or consistent information on who collected the data; the data have been recorded by staff members at TCFD over periods of changing protocols, and we have endeavored the best we can to interpret notes, recordings, and missing data entries. However, this aspect of the study is also of importance; the data are collected as individuals go about their daily lives over an extended period of time. Compared to a clinical trial with strict data compliance, we are able to retain more individuals and train the models to predict outcomes in their daily lives.

It also should be noted that many of the individuals had illnesses or started new medications during the data collection periods, which could impact their behavior. Side effects of these medications could also impact sleep or gastrointestinal issues. Due to the wide range of side effects, diversity of medications taken by individuals, and the lack of precise dates for some of the medications, these data were not included in the analysis. Additionally, data collection spanned before and after the beginning of the COVID-19 pandemic, which encompassed some large changes in environment and socialization for these individuals. Another feature set that should be examined in future work would be the use of FBAs for each individual and the observed function for each behavior occurrence. Challenging behaviors can occur for many reasons and connecting the physiological parameters we examined here with the psychological functions of the behavior may help to better serve the needs of these individuals.

Additionally, some variables selected for inclusion in our analysis have more supporting literature than others. For example, immune, GI, and sleep problems are common in the population with ASD, and sleep has been used to predict behavior [13]. However, weather and lunar data have less supporting literature. We leveraged sleep and GI data, which were already collected at the residential facility, then selected several data categories that had not been previously examined in relation to challenging behavior, and that could be collected from reputable outside sources. 

## 5. Conclusions

This study utilized longitudinal data collected from a residential facility for individuals with ASD to investigate correlations with and make predictions about behaviors from medical and environmental variables. The models had to be developed for each individual based upon past information about GI and sleep variables, as well as environmental variables. It was found that the GI variables were consistently the most important for individual behavior prediction. Surprisingly, sleep and allergen variables were important for some individuals but not all of them. Using the developed models, it was possible to predict with up to 90% accuracy if an individual would be likely to have a behavioral episode the next day. In general, the models of individuals with less class imbalance, i.e., they exhibited challenging behaviors on some days but not all, resulted in more accurate predictions. Future work will examine how to identify individuals whose behaviors can best be predicted by models, as well as carry out validation of the developed models on data collected in the future. 

## Figures and Tables

**Figure 1 jpm-13-01513-f001:**
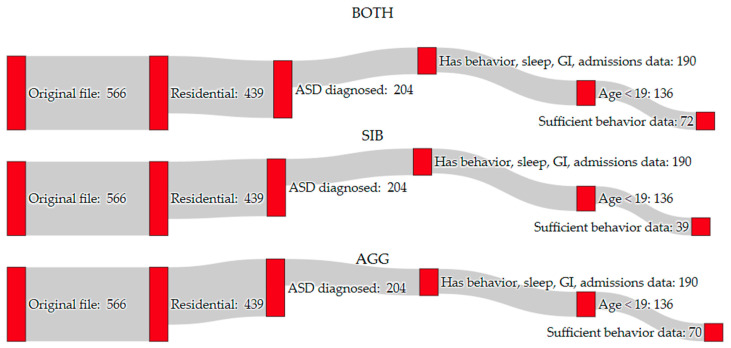
Inclusion criteria for each behavior cohort of the study (aggressive behavior—AGG; self-injurious behavior—SIB; BOTH refers to AGG and SIB). A formal ASD diagnosis upon entry to the center excluded the most individuals, while time periods with little data also excluded a high number of individuals. Allowing either behavior provided more data for individuals, so fewer participants were eliminated in the BOTH cohort, while the smallest cohort was the SIB group. However, individuals with one behavior that was much more frequent than the other were excluded from the BOTH cohort when there were few examples of no behavior.

**Figure 2 jpm-13-01513-f002:**
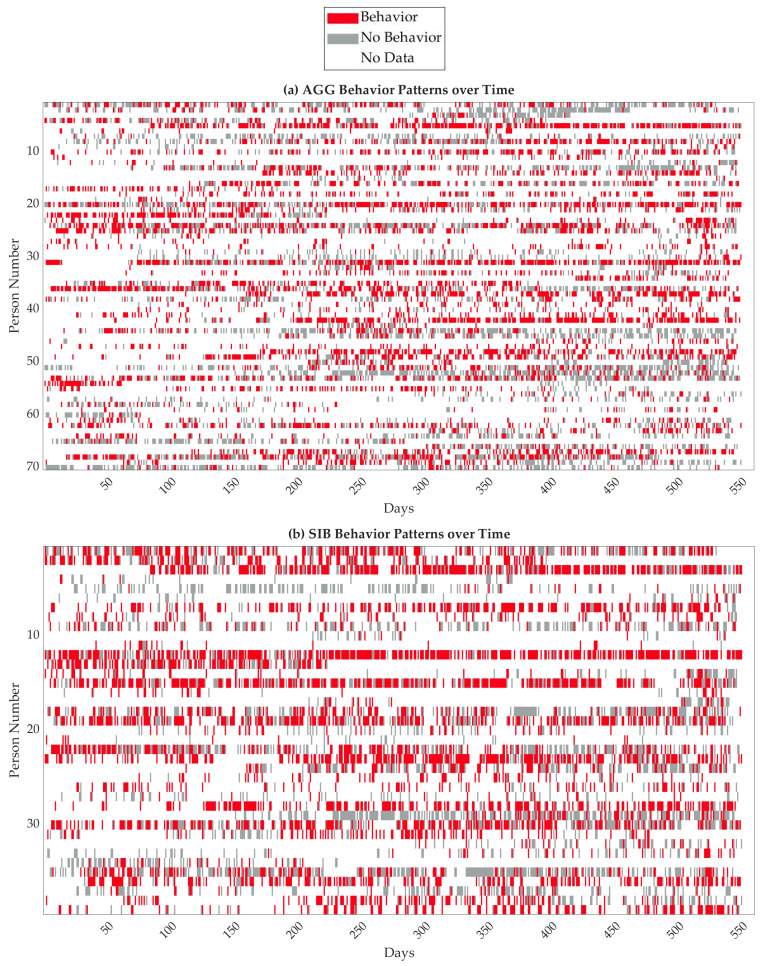
The filtered behavior patterns over time for each individual and for all behaviors examined. Behavior patterns differ largely; while some people typically have a behavior on most days, others rarely exhibit behaviors. Thus, class imbalance is of concern when predicting individuals’ behaviors. Subfigure (**a**) refers to individuals with aggressive behavior, (**b**) refers to individuals with SIB, and (**c**) refers to individuals with either aggressive behavior or SIB.

**Figure 3 jpm-13-01513-f003:**
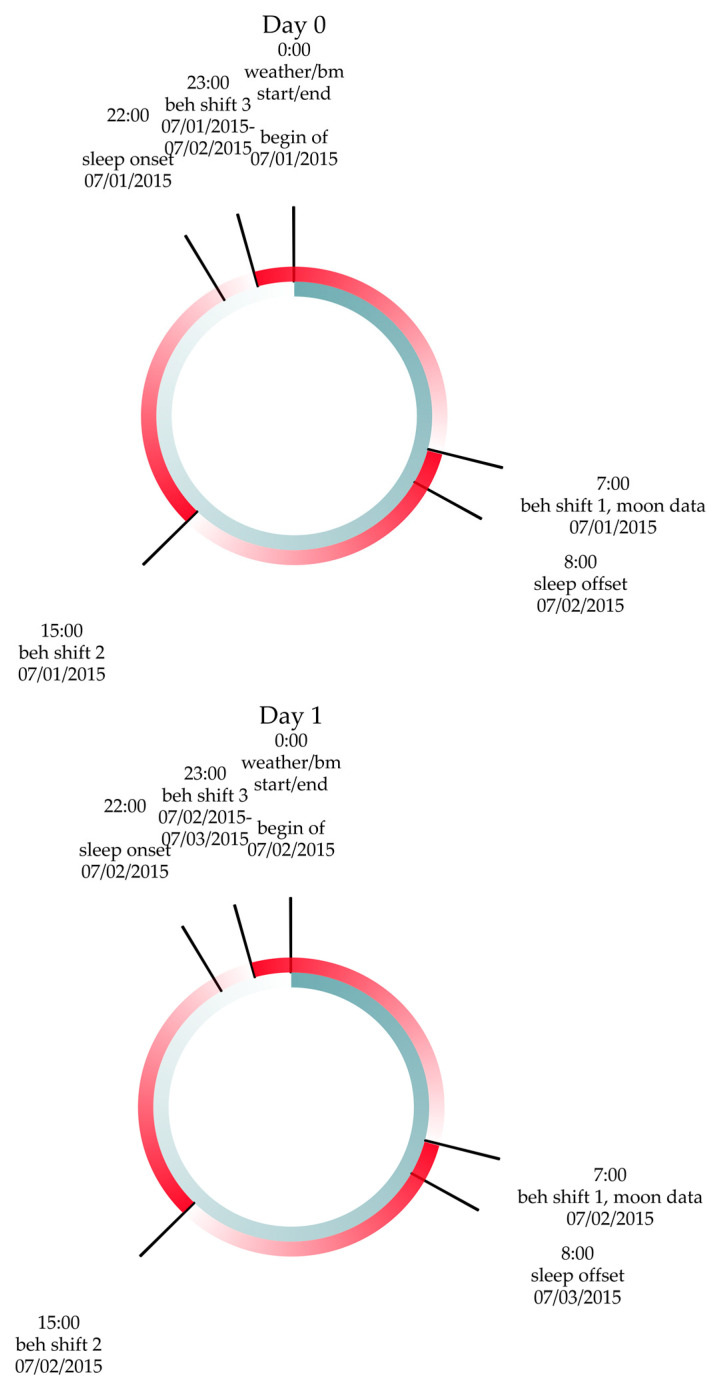
Times of data collection from Day 0 to Day 1, starting on 07/01/2015 as an example. The behavior day starts at 7 a.m., while the weather and BM data are both defined from midnight to midnight. While the moon data also has midnight recorded as a day start, the moon data were taken at 7 a.m. to align with the behavior data collection start. The behavior data from 07/01 7 a.m.–07/02 7 a.m., sleep data starting the evening of 07/01 to morning of 07/02, BM data starting at midnight 07/01 to midnight 07/02, moon data midnight 07/01 to midnight 07/02 collected at 7 a.m. 07/01, and weather data midnight 07/01 to midnight 07/02 are used to predict behavior 07/02–07/03 starting at 7 a.m.

**Figure 4 jpm-13-01513-f004:**
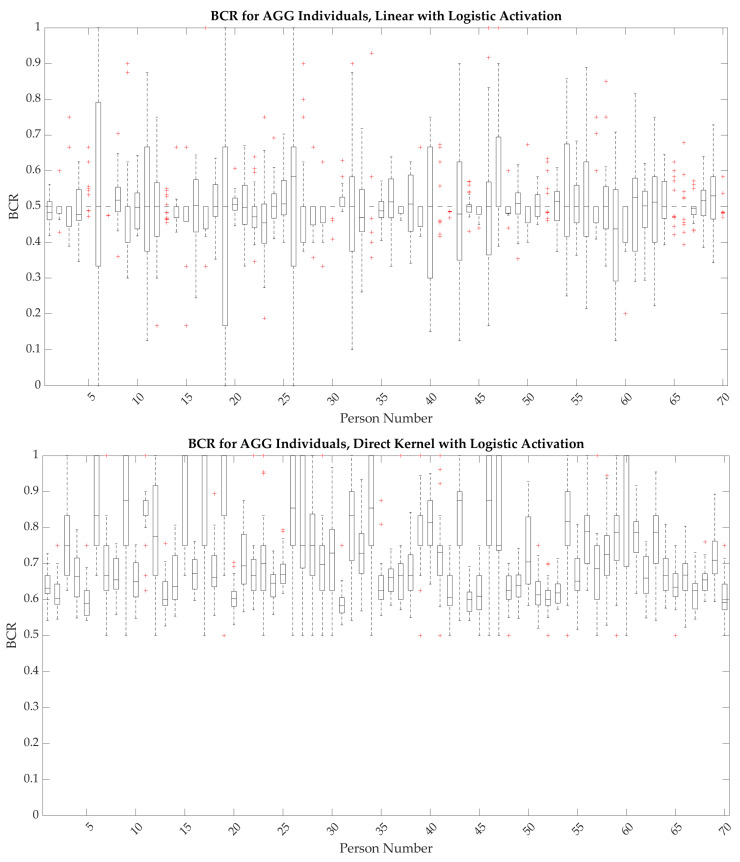
Linear results followed by direct kernel results for all individuals and cohorts: AGG at the top, SIB in the middle, and BOTH at the bottom. For all cohorts, the median linear model performance is close to 50%, indicating that other methods must be explored to achieve useful predictions. In almost all cases, the median BCR for the direct kernel approach is above 50%, and for many individuals the predictions are substantially better.

**Figure 5 jpm-13-01513-f005:**
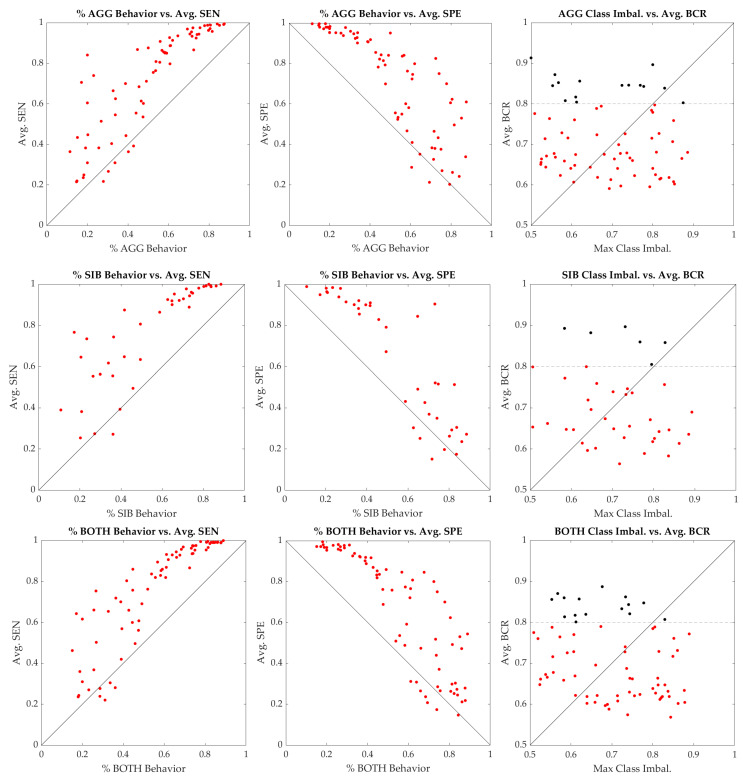
The sensitivity (**left**), specificity (**middle**), and BCR (**right**) for each model compared to the percentage of behavior occurring in the case of sensitivity and specificity, and the maximum class imbalance for BCR. Individuals with an average BCR above 80% are shown in black. All three cohorts (AGG on top, SIB in the middle, and BOTH at the bottom) follow similar trends for sensitivity, specificity, and balanced accuracy. Individuals with a smaller amount of class imbalance, i.e., they either exhibit a somewhat similar number of days with as well as without behaviors and show a higher BCR relative to their average class imbalance, while individuals with a higher amount of class imbalance are typically predicted poorly.

**Figure 6 jpm-13-01513-f006:**
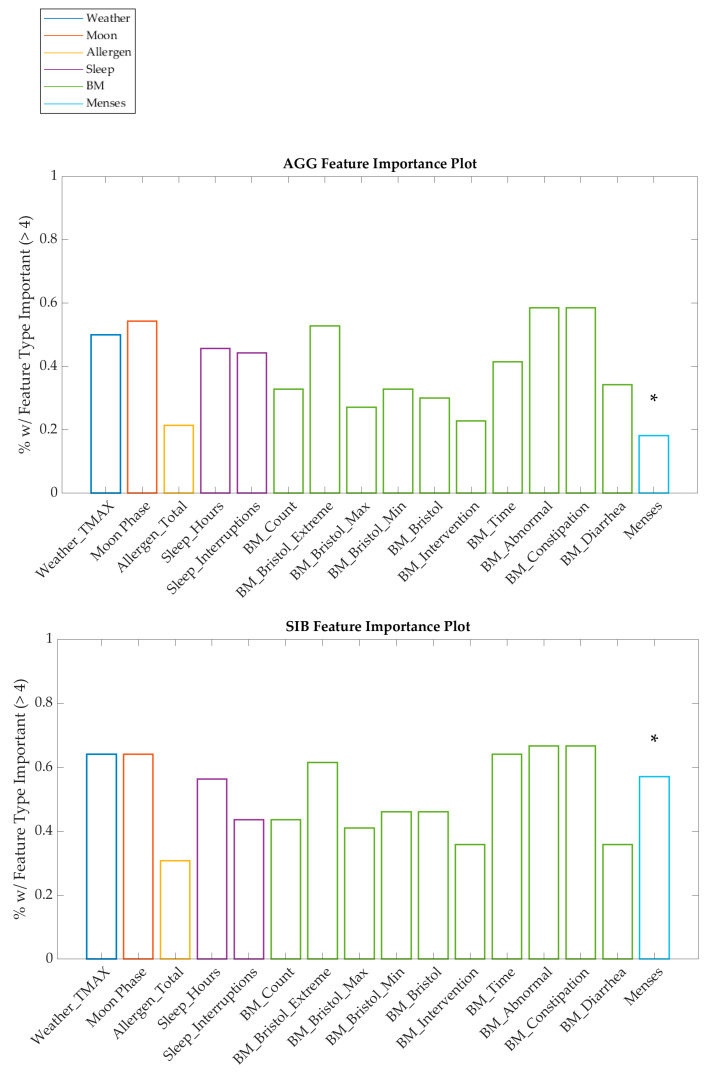
Plots of feature importance across all individuals show that BM, moon, sleep, and weather features are represented in the sensitivity analysis for the most individuals, while the allergen and menses features are less represented, except in the case of menses for the SIB cohort. Note that the results for menses, marked with an asterisk, are only calculated for the number of female individuals in each cohort.

**Figure 7 jpm-13-01513-f007:**
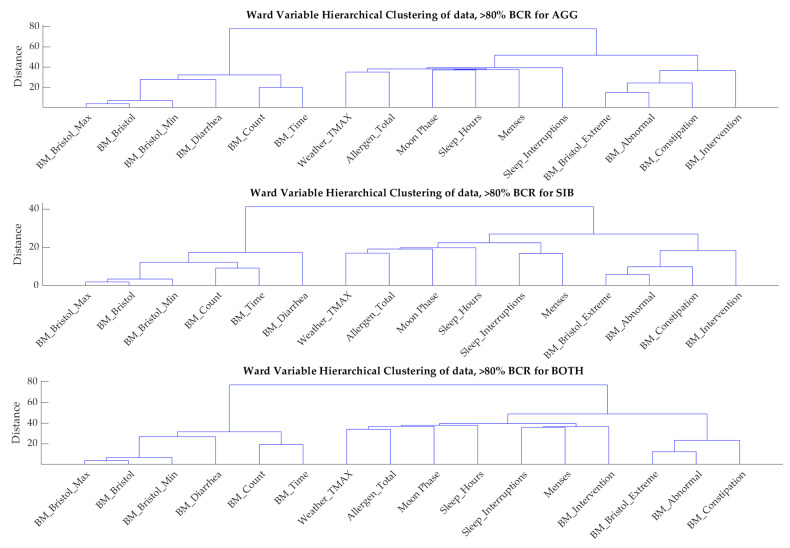
Hierarchical clustering of the data for the individuals where the balanced accuracy exceeds 80%. The clustering results for the entire population are very similar to the one for the individuals where high prediction accuracy was possible (results not shown).

**Table 1 jpm-13-01513-t001:** Individual demographics of the sample.

Cohort	Sex (Male; Female)	Age Range
AGG	59; 11	9.63–18.81
SIB	32; 7	11.00–18.81
BOTH	59; 13	9.63–18.81

## Data Availability

The de-identified data files used for this work can be downloaded at https://github.rpi.edu/ferinj/jpm-behavior.

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
