# Peer review of "Predicting Problematic Behavior in Autism Spectrum Disorder Using Medical History and Environmental Data"

_jpm, 2023, doi:10.3390/jpm13101513_

Round 1

Reviewer 1 Report

This study investigates how use artificial intelligence models to  predict the occurrence of aggressive behavior and self-injurious behavior in autism spectrum disorder population. The ability of predict problem behaviors is very important since strategies can be applied to avoid or mitigates them, with the very important effect of improving the quality of life of people with ASD, their families and more in general the society.

This work is interesting since the applied models are trained on a large set of data of 80 adolescent users ( < 19 years old at the study start time) in a residential setting. To enhance the data analysis and predictions authors exploit a detailed data set that combines daily clinical information of these patients with environmental data, over a long time period. Participants have been organized in three cohorts: aggressive, self-injurious and both behaviors.

Authors investigates if many variables (clinical data, sleep, gastrointestinal, allergen, environmental factors) can be predictors across the participants of problem behaviors.

Results indicate that the proposed models predict occurrences of behavior and non-behavior with high accuracies for some individuals in the cohort (a day before their expression). Models of participants showing problem behaviors on some days but not all, produced more accurate  predictions. Although the best prediction model varies from individual to individual, environmental and gastrointestinal factors were predictors across the participants.

This paper is fluent and easy to read. The related work is rich and critically discussed. The method is robust and illustrated in details, thus enabling reproducibility. Data are accurately analyzed and discussed. The discussion compares findings with previous work and suggest future investigation challenges. Limitations are clearly discussed.

It would be very valuable to share the data set.

IMO, the paper is ready to  be published.

Minors

Please explicit IRB acronym

Please explain the meaning of red vs grey segments (behavior vs non-behavior) in the behavior patterns in figure 2.

Reviewer 2 Report

In this manuscript of "Predicting Problematic Behavior in Autism Spectrum Disorder using Medical History and Environmental Data", the authors explored the use of artificial intelligence models to predict behavior episodes based on past data of co-occurring conditions and environmental factors for 80 individuals in a residential setting.

The manuscript is interesting, and well written. I suggest to accept it as this present version.

Reviewer 3 Report

JPM 2581373 Notes

I appreciate the opportunity to review and provide a recommendation on this manuscript, entitled “Predicting Problematic Behavior in Autism Spectrum Disorder using Medical History and Environmental Data”. As someone who provides clinical care and has engaged in research with the population of individuals on the autism spectrum with significant support needs and significant forms of challenging behavior, I am thrilled to see the authors focusing on this topic. Challenging behavior such as aggression and self-injury not only cause direct harm to these individuals and those who care for them, they substantially limit their autonomy and quality of life. Thus, efforts to understand relations between episodes of these forms of behavior and physiological or environmental variables has the potential to yield insights that could help to address them. I also have an appreciation for the challenges and opportunities that exist when examining data collected during the course of clinical care. For these reasons, I was eager to read this work.

Despite my enthusiasm for the general topic, I found that this study suffers from some significant limitations and that the means of presenting the rationale, methods, and findings in this manuscript did little to ameliorate those limitations. For this reason, my recommendation is that the authors’ request to have this work published in the Journal of Personalized Medicine be rejected. I will dedicate the remainder of these comments to highlighting a few of the issues that most informed this recommendation:

The rationale for the study is not sufficiently well developed. Several of the reasons provided in the manuscript for why the ability to predict episodes of challenging behavior is of value are not clear. Furthermore, the manner in which this study extends or exceeds the utility of methods for prediction based on previous research are based on assertion, without supporting details. For example, the authors note that there the work of Goodwin and colleagues has shown that it is possible to predict an episode of challenging behavior within a few minutes, but state that “more long-term prediction is desirable” without explaining why. As clinician, I find the additional precision of being able to predict an episode of challenging behavior within a few minutes far more helpful that knowing whether challenging behavior may occur on a given day. Similarly, the manuscript acknowledges that biosensor and motions sensors, which have been shown to produce data that can predict behavioral episodes, are well tolerated, but “may not be applicable to all individuals with ASD and high support needs” without providing any explanation for how that is the case. Having done work in the area of using biosensors on youth with ASD and significant support needs who exhibit extremely severe forms of challenging behavior I was initially surprised at how consistently we are able to get them to habituate to wearing the sensors. Thus, if there is a reason why these authors feel that their methods improve on those using biosensors due to intolerability they need to share those details.

More significantly, the rationale for including certain variables in the predictive models evaluated were not particularly well explored, especially early in the manuscript when they are most needed by readers. The degree to which the reasoning behind including certain variables are included is quite variable, with some explanation for a few variables, such as gastrointestinal symptoms, and none whatsoever for others, such as the lunar cycle. The result is that readers are left with the impression that the authors are on a “fishing expedition” in which the data on these variables were what they had available, and were therefore included, rather than there being a clear hypothesis that informed selection of relevant variables and collection of the corresponding data. This issue is particularly important if the authors wish to include a variable such as lunar cycle, which is associated with a history of superstition.

When a rationale is provided for including a particular variable, it generally comes in the form of a brief review of prior evidence of some association, without much additional context, such as explanation of the potential mechanism. An exception is the inclusion of data on the weather, where the paper suggests that it could impact the routine of participants, such as by preventing their ability to go outside or disruption of their daily routine. However, given that these were residential patients for which an opportunity exists to monitor many facets of their activity, if this was the authors’ hypothesis why didn’t they instead simply document whether they went outside that day or had their routine disrupted? Similar examples of including data on higher level environmental data rather than documenting individual participant experiences resulting from those purported variables exist, such as with the inclusion of the allergen report.

The quality of the data is not reported. Given that these data were collected clinically, it seems highly likely that consistency and vigilance of those collecting data on challenging behavior or other participant behavior (e.g., sleep) varied significantly across data collectors. This is to be expected in a clinical setting, but some measure to ensure at least reliability would be important to include. For example, a second data collector for a percentage of observations would allow for a calculation of interobserver agreement.

I strongly encourage the authors to delve more deeply into the literature on functional behavioral assessment. They hint at this literature somewhat, by referencing the work of Iwata and colleagues. However, there is substantial research into the potential etiology of challenging behavior in this population, including the development of a relatively robust categorization system based on the function of the behavior for the individual participant. This system defines challenging behavior as either socially mediated (i.e., maintained by extrinsic consequences that involve another person) or automatically maintained (i.e., maintained by the consequences of the behavior itself – often [although not always] through providing or attenuating sensory stimuli). Several prior studies, such as the work of Kennedy or Hagopian and colleagues, have examined how the function of challenging behavior may moderate the effects of environmental or physiological events, such as headaches, constipation, etc. Including the results of a functional assessment in these analyses would make the present study much more interesting.

The results of the study are also not particularly strong with the proportion of individuals who met the predetermined threshold for ability to predict a day with challenging behavior was ~15-20%. Such a result is unlikely to be of much utility to readers.

Finally, the manuscript states that IRB approval was not required. Although IRB approval may not be required to render clinical care to participants, at my institution even a retrospective chart review leading to dissemination of participant data requires IRB approval.

Overall, the writing of the paper was fine, although there were several grammatical errors (e.g., using "while" instead of "although") etc.

Round 2

Reviewer 3 Report

I appreciate the authors’ attempts to address the concerns I raised in my prior review. They clearly worked rapidly to deal with as many as possible. However, I regret that these revisions are likely inadequate to raise the level of impact of this work to meet the bar for publication in this outlet. Specific lingering issues include the following:

One issue I noted was that the rationale had to do with the clinical significance of predicting whether challenging behavior would occur on a particular day. The authors attempted to address this criticism by adding a paragraph explaining how being able to predict whether challenging behavior would occur on a given day allows caregivers to deploy “countermeasures”, such as rearranging the schedule for the day, increasing staffing ratios, etc. However, if the authors read my critique of this part of the rationale more carefully, they will see that my concern is not that being able to predict an episode of challenging behavior is not clinically useful. Rather, my critique had to do with the fact that other researchers have already done work using biosensors that allow for prediction of episodes of challenging behavior within a few minutes. Thus, the need for revision centers around why prediction at the level of an entire day is more useful than the existing studies that provide much more precision. The authors go on to state that care facilities would derive greater benefit from less precise predictions because of the ability to adjust staffing levels. However, as someone who oversees a large care facility, I can say with some assurance that I find much greater value in being able to flex staff momentarily based on a need to address challenging behavior rather than having to increase or decrease staff for an entire day, which is frankly far less financially sustainable.

I appreciate that the authors included more background on functional assessment. However, it still did not make note of the literature that evaluated the relationship between function of an individual’s challenging behavior and physiological variables, which would be particularly relevant to this study. Furthermore, my original intent with my prior review was to note that inclusion of information on the function of individual participants’ behavior in combination with the data that were included would make for a far more interesting paper. I suspect that those data are perhaps unavailable or they would have been included in the original analysis. Thus, including additional background on functional assessment is helpful, but absent additional data, the impact of this analysis is diminished.

The authors made a good faith effort to address my concern that the rationale for inclusion of several variables in their analysis was not strong. Text has been added that provide citations of more studies that suggest relationships between some of these variables and challenging behavior. However, the literature is not equally strong for each of those relations with challenging behavior. Furthermore, part of my original critique was that the purported mechanism accounting for relations between some of these variables and challenging behavior seem to be higher order, when the ability to simply collect data on the more direct mechanism would have been better. In my review I mentioned that the hypothesis for the relationship between the weather and challenging behavior included the fact that weather events could affect whether participants were able to go outside or could disrupt their schedules. However, if that is the hypothesis, why not just document whether participants were able to go outside, or had their schedules disrupted. This issue is highlighted by the changes made to this version of the manuscript to give a rationale for inclusion of lunar cycle in the analysis. The paper now states that lunar cycle may be related to sleep. Yet the analysis already includes sleep as a variable. Thus, this rationale not only doesn’t solve the issue raised, but highlights the concern.

Finally, my prior review also raised the concern that there was not safeguards for data integrity/fidelity, such as gathering interobserver agreement data. This issue remains unaddressed, and probably cannot be with the current dataset.

Again, I appreciate the authors’ attempts to bring attention to the serious difficulties and dangers raised by challenging behavior and its impact on quality of life for individuals with autism. Unfortunately, I do not believe that the current manuscript is publishable within this outlet. Furthermore, I do not believe that the issues that diminish its likely impact on the literature are amenable to resolution through additional revision. Despite this, the authors have respect for working on this topic. I wish them the best in their future scholarly endeavors, and hope that they will consider this journal as a potential outlet for their work.

Author Response

Please see the included cover letter for a detailed response to the reviewer's comments.

As far as changes to the paper are concerned, we have provided a limitations section that clearly lists the limitations of this work as suggested by the academic editor.

That being said, the authors do not believe that sending the paper back to the reviewer will result in a change of the outcome (as the reviewer clearly states that the paper cannot be made publishable in his/her opinion).